# CD200R1 Contributes to Successful Functional Reinnervation after a Sciatic Nerve Injury

**DOI:** 10.3390/cells11111786

**Published:** 2022-05-30

**Authors:** Bruno Pannunzio, Jesús Amo-Aparicio, Camila Julián, Rubèn López-Vales, Hugo Peluffo, Natalia Lago

**Affiliations:** 1Neuroinflammation and Gene Therapy Laboratory, Institut Pasteur de Montevideo, Montevideo 11400, Uruguay; bpannunzio@pasteur.edu.uy (B.P.); hpeluffo@pasteur.edu.uy (H.P.); 2Departamento de Histología y Embriología, Facultad de Medicina, Universidad de la República, Montevideo 11800, Uruguay; camilajulian21@hotmail.es; 3Departament de Biologia Cel·lular, Fisiologia i Immunologia, Institut de Neurociències, Centro de Investigación Biomédica en Red sobre Enfermedades Neurodegenerativas (CIBERNED), Universitat Autònoma de Barcelona, 08193 Bellaterra, Catalonia, Spain; jesusamoaparicio@gmail.com (J.A.-A.); ruben.lopez@uab.cat (R.L.-V.); 4Unitat de Bioquímica i Biologia Molecular, Departament de Biomedicina, Facultat de Medicina i Ciències de la Salut, Universitat de Barcelona, 08036 Barcelona, Catalonia, Spain

**Keywords:** CD200R1 blocking antibody, immune receptor, macrophages, regeneration, functional reinnervation, Wallerian Degeneration

## Abstract

Activating and inhibitory immune receptors play a critical role in regulating systemic and central nervous system (CNS) immune and inflammatory processes. The CD200R1 immunoreceptor induces a restraining signal modulating inflammation and phagocytosis in the CNS under different inflammatory conditions. However, it remains unknown whether CD200R1 has a role in modulating the inflammatory response after a peripheral nerve injury, an essential component of the successful regeneration. Expression of CD200R1 and its ligand CD200 was analyzed during homeostasis and after a sciatic nerve crush injury in C57Bl/6 mice. The role of CD200R1 in Wallerian Degeneration (WD) and nerve regeneration was studied using a specific antibody against CD200R1 injected into the nerve at the time of injury. We found an upregulation of CD200R1 mRNA after injury whereas CD200 was downregulated acutely after nerve injury. Blockade of CD200R1 significantly reduced the acute entrance of both neutrophils and monocytes from blood after nerve injury. When long term regeneration and functional recovery were evaluated, we found that blockade of CD200R1 had a significant effect impairing the spontaneous functional recovery. Taken together, these results show that CD200R1 has a role in mounting a successful acute inflammatory reaction after injury, and contributes to an effective functional recovery.

## 1. Introduction

After a peripheral nerve injury, the distal portion of the nerve undergoes progressive degeneration in a process known as Wallerian Degeneration (WD) (Waller 1850). WD begins with axonal degeneration, followed by myelin ovoid breakdown and myelin clearance by Schwann cells and resident and infiltrating macrophages and neutrophils [1,2], taking 14 to 21 days to clear axonal and myelin debris. Rapid WD results in an extracellular environment that promotes axon regeneration, thus problems in monocyte recruitment, a chronic inflammatory response and deficits in myelin and debris clearance could all create an inhibitory environment for axonal regeneration. While WD in the peripheral nervous system (PNS) is perfectly orchestrated, in the central nervous system (CNS) slow WD results in the prolonged presence of myelin-associated inhibitors that likely contribute to the failure of CNS axon regeneration [3,4,5]. Thus, endogenous molecular mechanisms or therapeutic compounds increasing the speed of WD and regulating the rapid activation and resolution of inflammation might enhance axonal regeneration and reinnervation. 

Following injury to the nerve, Schwann cells (SCs) undergo a dramatic reprogramming from highly quiescent to a proliferative, pro-repair state [6]. Dedifferentiated SCs have different roles in orchestrating the regeneration process. Very early after injury, the axonal breakdown is initiated and products of degenerated neural tissue stimulate SCs and resident macrophages to secrete chemokines and cytokines, which play an important role in the recruitment of macrophages and neutrophils from circulation and in their activation [1,4]. Accordingly, a few hours after nerve injury, neutrophils enter the nerve and, together with SCs, contribute to the axonal breakdown and phagocytosis of the axonal and myelin debris [7,8,9]. Therefore, apart from SCs and resident macrophages, infiltrating monocytes and neutrophils have a core role first in myelin and debris clearance and after in regeneration [10,11,12]. These three cell populations work in this way in an orchestrated manner after peripheral nerve injury [7], and thus, the response of the immune system has been shown to be central in exhibiting a successful regeneration and reinnervation after an injury to the PNS. In this sense, searching for immune checkpoint mechanisms could contribute towards elucidating the role of the inflammatory response after injury to the nervous system and point towards novel therapeutic targets. 

Activating and inhibitory immune receptors, such as CD300f, TREM2, CD200R1 and Siglecs, have been shown to mediate important functions in the modulation of neuroinflammation [13,14]. For instance, CD300f, a dual activating and inhibitory immune receptor, has been shown to be implicated in WD and nerve regeneration after a nerve crush by modulating the influx and phenotype of macrophages and stimulating regeneration [14]. CD200R1 has been proposed as a negative regulator of microglia by interacting with its ligand CD200 under different inflammatory conditions of the CNS [13,15]. There are four CD200Rs (1–4) with activating or inhibiting potential, however, only CD200R1 has been shown to engage CD200 [16]. The putative anti-inflammatory function of CD200R1 is based on the fact that mice lacking CD200, which is expressed in several cell types including neurons, astrocytes and endothelial cells [13,17,18], display worsened clinical scores in a model of experimental autoimmune encephalitis (EAE), more microgliosis after facial nerve transection [13], impairment in long-term potentiation and increased numbers of microglial cells in naive animals [18], and increased inflammation and functional deficits after spinal cord injury [19]. On the other hand, few studies have studied the role of the pair CD200-CD200R1 by manipulating the receptor. Mice lacking CD200R1 showed an increased peripheral inflammation with higher mortality rates and increased peripheral recruitment of myeloid cells after ischemic stroke [20], and blocking CD200R1 induced an increased inflammatory milieu and worsened functional deficits after spinal cord injury [21], supporting the importance of CD200R1 in preventing inflammation after CNS trauma. However, no data is available regarding CD200R1 and its putative role in WD and functional recovery after nerve injury. 

In the present study, we first characterized the expression of the pair CD200-CD200R1 in the sciatic nerve in homeostasis and after crush injury. Moreover, by using a well characterized specific blocking antibody against CD200R1, we investigated the effect of blocking the activation of CD200R1 after nerve injury. We found that blockade of CD200R1 impacts macrophage and neutrophil recruitment and impairs functional reinnervation.

## 2. Materials and Methods

### 2.1. Animal Surgery and Treatment

All experimental procedures were approved by the Institut Pasteur de Montevideo Animal Care Committee (approval number 005-19) and conducted according to international FELASA guidelines, national law, ethical guidelines (Uruguayan Animal Care Committee) and the ARRIVE guidelines for reporting animal research [22]. Adult (10–12 weeks old) male and female C57BL/6 mice weighing 25–30 g were used in this study. CD200R1 wild type (WT) and CD200R1 knock-out (KO) (B6N.129S5-Cd200r1^tm1Lex^/Mmucd) mice were purchased from Mutant Mouse Resource and Research Centers (UC Davis; Davis, CA, USA). Mice were maintained in a controlled environment (12-h light-dark cycle; 20 ± 1 °C and housed a maximum of 6 mice per cage. Surgical procedure was performed in the right sciatic nerve of anesthetized mice (Ketamine-Xylacine; 90–10 mg/Kg; ip). The nerve was crushed at about 45 mm from the tip of the third digit for 30 s with fine forceps. Injections were performed directly in the sciatic nerve prior to the crush at the exact same position of the injury. Then, 2 µL containing 2 μg of rat anti-mouse CD200R1 (Clone OX-131; FcSilentTM Absolute Antibody; Oxford, UK) or rat-IgG1 (FcSilentTM Absolute Antibody; Oxford, UK) were injected using a Hamilton 33-gauge needle connected to a 10 µL Hamilton syringe. The wound was closed and disinfected afterwards.

All experiments were conducted in C57BL/6 male mice, with the exception of flow cytometry analyses where female mice were used. No animals were excluded from the different studies. 

### 2.2. Histological and Immunohistochemical Procedures

Mice were deeply anesthetized with pentobarbitone at 1, 3, 7, 14 and 28 dpi and intracardially perfused with 4% paraformaldehyde (PFA; Sigma; St. Louis, MO, USA) in 0.1 M phosphate buffer solution (Sigma, St. Louis, MO, USA). The sciatic nerve was dissected to the ankle level and harvested. Tissue was postfixed in 4% paraformaldehyde for 3 h, transferred to 30% sucrose (Sigma; St. Louis, MO, USA), and frozen for further immunohistochemistry procedures. The tibial nerve at the ankle level was dissected out, postfixed in 2% glutaraldehyde in 0.1 M phosphate buffer, and processed for embedding in Epon resin for semithin section preparations. 

Longitudinal sections (8 μm thickness) of the sciatic nerve were taken and stored at −20 °C until used. Non-specific antibody binding was blocked with PBS 0.01 M + 1% Triton + 10% fetal bovine serum (FBS; GIBCO 10270-106; Paisley, UK), for 1 h at room temperature. Sections were then incubated overnight at room temperature with the following primary antibodies: rat anti-mouse CD200 (1:100; Serotec MCA1958; Oxford, UK), rabbit anti-S100β (1:150; Sigma SAB5700647; St. Louis, MO, USA), rat anti-Ly6G (1:200 Biolegend 127601; San Diego, CA, USA), rat anti-PGP9.5 (1:500; Cedarlane CL7756AP; Burlington, ON, Canada) and lectin from *Lycopersicon esculentum* (1:100 Sigma L9389; St. Louis, MO, USA). After washes with PBS-Triton 1%, sections were incubated for detection with appropriate secondary antibodies or Streptavidin (Invitrogen; Waltham, MA, USA) and DAPI (D9542; Sigma; St. Louis, MO, USA). Samples incubated without the primary antibody were included as controls for nonspecific binding. 

For immunofluorescence of teased fibers, sciatic nerves were freshly dissected out and immediately immersed in 4% paraformaldehyde in 0.1 M phosphate buffer for 3 h. After washing with PBS, the perineural sheath was removed and nerve bundles were separated using a pair of fine needles. Teased fibers were blocked and then incubated with the following primary antibodies overnight at room temperature. After washes with PBS- Triton 1%, sections were incubated for detection with appropriate secondary antibodies (Invitrogen; Waltham, MA, USA) and DAPI. Confocal images of teased fibers were acquired using a ZEISS LSM 880 confocal microscope.

Skin innervation was evaluated by analyzing the density of nerve fibers present in the epidermis of the hind-paws. Plantar pads were dissected out at 28 dpi, cryopreserved and processed as described [14]. Blockade of non-specific antibody binding was conducted in 70-μm cryostat sections with PBS 0.01 M + 0.3% Triton + 1% normal goat serum for 1 h at room temperature. Sections were then incubated in primary rabbit antibody against protein gene product 9.5 (PGP9.5, 1:500; Cedarlane CL7756AP; Burlington, ON, Canada) for 24 h at 4 °C and with a donkey anti-rabbit Cy3 (1:200; Millipore; Darmstadt, Germany) for 24 h at 4 °C, and mounted on gelatin-coated slides. Five sections from each sample were used to quantify the nerve fibers present in the epidermis of the paw pads. Tissue sections were examined using an OlympusIX81 microscope and images of the longitudinal sections were acquired at 20× with an AxioCam MRm Zeiss camera attached to a computer for further counts and imaging processing by using ImageJ software. 

The total number of neutrophils in sciatic nerve sections was determined at 1 dpi by counting the number of cells in the total slide. After incubation with anti-Ly6G primary and secondary antibody, epifluorescence images of the whole sections were acquired using a ZEISS LSM 800 microscope and tiled images were stitched together. Three slides per animal were used to quantify the number of Ly6G positive cells. Cell count was performed using Image J software and normalized by the total area of the sciatic nerve section. 

Semithin sections (1 μm) were obtained at 28 dpi from the tibial nerve blocks. Images of whole tibial nerve cross section were acquired at 10× with a Media Cybernetics PL-A662 camera attached to a computer, while sets of images chosen by systematic random sampling of squares representing at least 30% of the nerve cross-sectional area were acquired at 100×. Measurements of the cross-sectional area of the whole nerve, as well as counts of the number of myelinated fibers, were carried out using ImageJ software.

For myelin clearance analyses, 8 μm cryostat longitudinal nerve sections were stained with Luxol Fast Blue (LFB; Sigma; St. Louis, MO, USA). After graded dehydration, sections were placed in a 1 mg/mL LFB solution in 95% ethanol and 0.05% acetic acid overnight at 37 °C. Sections were then washed in distilled water before being de-stained in a solution of 0.05% Li_2_CO_3_ in distilled water for 10 s and given a brief rinse in 70% Ethanol. Sections were then dehydrated and mounted in DPX mounting media (Sigma; St. Louis, MO, USA). For degenerated myelin analyses, sections were stained with Oil Red O (ORO, Sigma; St. Louis, MO, USA). Sections were incubated in ORO solution for 10 min at room temperature and then placed under running tap water for 30 min. Sections were then mounted on 80% glycerol. Images were acquired at 20× with an Olympus CX41 microscope coupled with a Media Cybernetics PL-A662 camera. Measurements of LFB or ORO positive areas were performed using ImageJ software. 

### 2.3. Flow Cytometry

Naïve and crushed injured mice at different time points were perfused with ice-cold PBS to remove blood assuring that cells evaluated are from the neural parenchyma. Single sciatic nerves were harvested and cut into pieces before the enzymatic digestion in 0.125% collagenase and 0.125% DNAse at 37 °C for 30 min. Digested tissue was then filtered through a 30 μm cell strainer. After centrifugation, cells were resuspended and incubated for 1 h at room temperature with combinations of the following antibodies: anti-CD45 PerCP (1:100 BD Pharmingen #557235; San Diego, CA, USA), anti-CD11b PECy7 (1:100, Biolegend #101216; San Diego, CA, USA), anti-F4/80 APC (1:100, eBioscience #17-4801-82; San Diego, CA, USA), anti-Ly6G PE (1:100, BD Pharmingen #551461; San Diego, CA, USA), anti-Ly6C FITC (1:100, BD Pharmingen #553104; San Diego, CA, USA), anti-CD206 FITC (1:100, Biolegend #141704; San Diego, CA, USA), anti-CD16/32 PE (1:100 Biolegend #101308; San Diego, CA, USA) and anti-CD200R1 Alexa Fluor 647 (1:100, BD Pharmingen #566345; San Diego, CA, USA). After incubation and a washing step, samples were analyzed with a BD FACSCanto II Flow Cytometer or an Attune NxT Flow Cytometer. FlowJo Software (BD Bioscience; Franklin Lakes, NJ, USA) was used to further analyze the results. 

### 2.4. Isolation of RNA and QPCR

Animals at different time points were perfused with ice-cold PBS and sciatic nerve were harvested (from 1 mm proximal to 6 mm distal to the crush). RNA was isolated and purified from homogenized nerves as described in [14] in TRIzol (SIGMA, T9424; St. Louis, MO, USA), and the aqueous phase was further purified using the Direct-zol RNA MiniPrep Plus kit (Zymo Research, R2070; Irvine, CA, USA). RNA samples were reverse transcribed using M-MLV reverse transcriptase (Invitrogen 28025-013; Waltham, MA, USA) and random primers. Quantitative PCR (QPCR) was performed using the following TaqMan reagents from Invitrogen/Applied Biosystems (Waltham, MA, USA): TaqMan Fast Advanced Master Mix (4444557), IL-1β (Mm01336189_m1), IL-10 (Mm01288386_m1), IL-6 (Mm00446190_m1), TNFα (Mm00443258_m1), CD68 (Mm00444543_m1), TGFβ (Mm00178820_m1), INFγ (Mm001168134_m1), CCL2 (Mm00441242_m1) and CCL3 (00441259_g1). The relative expression ratio is calculated using the real-time PCR efficiencies and the crossing point deviation of an unknown sample versus a control. GAPDH endogenous control (Mm99999915_g1) was included in the model to standardize each reaction run with respect to RNA integrity and sample loading. QPCR was performed using the QuantStudio 3 apparatus and software. Cycling conditions were 50 °C for 2 min, 95 °C for 2 min, followed by 40 cycles at 95 °C for 1 s and 60 °C for 20 s.

### 2.5. Functional Evaluation

The walking track sciatic functional index (SFI) was carried out to assess recovery of locomotor function. Prior to lesion and at 1, 3, 5, 7, 10, 14, 17, 21 and 28 days following crush injury, mice were allowed to walk down a runway after inking the plantar surface of their hind-paws. The print length (PL) and the distance between the first and fifth toes (toe spread, TS) and between the second and forth toes (intermediate toe spread, IT) were measured. The three parameters were combined in the SFI to quantify changes in walking patterns [23]. A researcher blinded to the treatment groups conducted all evaluations.

### 2.6. Bone Marrow-Derived Macrophage (BMDM) Generation and Treatment

BMDMs were isolated from 20-week-old CD200R1-WT and CD200R1-KO male mice as previously described [24]. Briefly, bone marrow cells were collected by flushing the femur and tibia shafts with DMEM/F12 supplemented with 10% FBS and Penicillin/Streptomycin (complete DMEM/F12). Cells were cultured for five days in 100 mm Petri dish with complete DMEM/F12 and 20 ng/mL recombinant mouse macrophage colony-stimulating factor protein (M-CSF, Biolegend #576406; San Diego, CA, USA). Cells were subcultured in 24-well cell culture plates in the same medium for 3 days. BMDMs were treated for 24 h with 500 ng/mL Lipopolysaccharide (LPS, D9542; Sigma; St. Louis, MO, USA) or rat anti-mouse CD200R1 (Clone OX-131; FcSilentTM Absolute Antibody; Oxford, UK, 5 µg/mL) in complete DMEM/F12 and 5 ng/mL M-CSF. After treatment, cells were resuspended in TRIzol and RNA was isolated as described previously.

### 2.7. Data Processing and Statistical Analysis

All data are shown as mean ± standard error of the mean (SEM). Statistical analysis of behavioral data (SFI) was determined using two-way repeated measures ANOVA followed by Bonferroni post-hoc analysis. One-way analysis of variance (ANOVA) followed by Tukey’s post-hoc analysis was used for experimental data with more than two experimental groups and normal distribution. For QPCR analysis, non-parametrical test, Kruskal–Wallis test followed by Dunn’s post-hoc analysis, were used. For comparison of the differences in the means of the two groups, a two-tailed unpaired Student’s t test was used. A value of *p* ≤ 0.05 was considered statistically significant. All analyses were performed by researchers blinded to the treatment.

## 3. Results

### 3.1. CD200-CD200R1 Expression in Sciatic Nerve

The expression of CD200R1 and its most well-known ligand CD200 were assessed in the sciatic nerve in homeostasis and after a crush nerve injury. QPCR analysis revealed that CD200R1 mRNA was constitutively expressed at low levels in the sciatic nerve (Figure 1A). After a crush injury, CD200R1 transcripts were rapidly upregulated in the sciatic nerve from 24 h post-injury and remained at high levels until 28 dpi, the latest time point studied (Figure 1A). To further study which cell populations express CD200R1, we performed flow cytometry analysis of the uninjured sciatic nerve and at one, five and fourteen days after crush injury. Macrophages were identified as CD45^+^/CD11b^+^/F4/80^+^ whereas neutrophils were identified as a CD45^+^/CD11b^+^/F4/80^−^/Ly6G^+^ cells (Figure 1B). As expected, at 24 h after nerve injury, there was a significant infiltration of myeloid cells such as neutrophils and monocytes (Figure 1B). Macrophages increased in numbers peaking at 5 dpi and decreasing at 14 dpi, whereas neutrophils peaked for the first 24 h and rapidly decreased thereafter (Figure 1C). Expression of CD200R1 protein was found in both naïve and injured nerve. In naïve animals, around 60% of resident macrophages expressed CD200R1 and this percentage did not vary in the macrophage population after nerve injury (Figure 1D,E). As expected, neutrophils were not found in the sciatic nerve in naïve conditions. Twenty-four hours after nerve injury, nearly 10% of infiltrating neutrophils expressed CD200R1 (Figure 1F). Interestingly, the CD200R1 median fluorescence intensity (MFI) was considerably lower in macrophages after the lesion at all timepoints, showing that in fact CD200R1 is downregulated or, alternatively, that invading macrophages express less CD200R1 than endogenous macrophages. Thus, though the CD200R1 mRNA increases, this is due to the influx of CD200R1+ macrophages and not by CD200R1 overexpression (Figure 1G). In summary, macrophages were the main cell type in the sciatic nerve expressing this receptor in both naive conditions and after nerve injury.

The expression of CD200 in the sciatic nerve was assessed by immunohistochemistry and QPCR in the uninjured and in the crushed sciatic nerve. In the uninjured nerve, immunohistochemistry of longitudinal sections and teased fibers evidenced CD200 expression by Schwann cells in Schmidt–Lanterman incisures and Ranvier nodes (Figure 2A–I), as reported previously [25] and in blood vessels (Figure 2A–F, arrowhead).

We then evaluated the expression of CD200 at different time-points by QPCR after crush injury to the sciatic nerve. QPCR showed a tendency to decrease in CD200 mRNA expression at 24 h after crush injury and then show a progressive significant increase thereafter, peaking at 14 dpi (Figure 2J), which was confirmed by IHQ with an expression not restricted to the SCs domains (Figure 2K–N).

### 3.2. Blockade of CD200R1 Impairs Functional Recovery

CD200R1 has been proposed as a negative regulator of macrophages and microglia keeping them in a homeostatic state [15]. In order to explore whether CD200R1 participates in the physiopathology of the nerve injury, a well characterized recombinant rat anti-mouse CD200R1 (OX131, αCD200R1) blocking antibody or rat IgG1 (Vehicle) [21,26] were injected into the sciatic nerve immediately after a crush injury, and the recovery of function was evaluated by analyzing the hind-paw prints to obtain the Sciatic Functional Index (SFI). The OX131 αCD200R1 used in this study was originally generated by the group of Neil Barclay and has been shown to be specific for CD200R1 in C57BL/6 mice and prevent inhibitory signaling induced by CD200 [26]. In our hands, it stimulated LPS-induced IL-1β mRNA expression in WT but not CD200R1 knockout bone marrow-derived macrophages (BMDM), suggesting that this antibody is indeed blocking CD200R1 (Appendix A). In addition, this antibody has been engineered to minimize FcγR receptor binding and reduced complement binding.

As expected, after the crush injury, the SFI dropped to −95%, whereas it recovered by day 28 up to −10% with no significant differences when compared to pre-injury values (Figure 3A). After nerve injection of αCD200R1, we observed a significant impairment of function from 17 dpi that was maintained until 28 dpi, the last time point studied (Figure 3A), when compared with the vehicle group. In addition to the functional index, skin reinnervation and nerve regeneration were assessed. Distal reinnervation was assessed by counting the number of PGP9.5-positive fibers in the epidermis of a pad. The number of PGP9.5-positive fibers was significantly lower in both injured groups when compared with the contralateral skin at 28 dpi but no differences were seen between treatments (Figure 3B,C). As an index of regeneration, myelinated axons were counted in semithin sections from the distal part of the tibial nerve. The number of myelinated fibers at 28 dpi was significantly lower in both groups with crush in comparison with uninjured nerve. Again, no differences in the number of regenerated myelinated fibers were found between both injured groups independently of the treatment (Figure 3D,E).

All together these results suggest that an acute inhibition of CD200R1 after a sciatic nerve crush has long term effects in functional recovery without significant effects in other regenerative outcomes.

### 3.3. Blockade of CD200R1 Delays Monocyte Recruitment after Sciatic Nerve Injury

In order to understand the mechanism by which there is an impairment in the functional recovery after blocking CD200R1 function, we assessed the level of inflammatory markers and macrophage infiltration and phenotype. mRNA expression of different cytokines was analyzed by QPCR in naive and at one day post-injury nerve samples. After a crush injury, increased mRNA levels of *Il1b*, *Tnfa*, *Il6*, *Cd68*, *Il10* and *Tgfb* were observed in both αCD200R1 and vehicle treated groups when compared with the naive group (Figure 4A–F). Although treatment with αCD200R1 seemed to reduce the mRNA expression for most of the cytokines analyzed, no significant differences were observed between both groups.

The early inhibition of CD200R1 could influence the pattern of infiltration of myeloid cells. This putative effect was assessed by flow cytometry analysis at one day post-injury. The total number of infiltrated CD45^+^/CD11b^+^/F4/80^+^ macrophages was significantly lower in animals injected with the αCD200R1 when compared to those injected with control IgG1 (Figure 4G,H). To further understand the effect that the inhibition in CD200R1 could have in macrophages, we evaluated two proinflammatory markers such as CD16/32 and Ly6C, and an anti-inflammatory marker such as CD206 [27]. Most of the macrophages present in the injured nerve were Ly6C^+^ and CD16/32^+^. Although the percentage of most of these macrophages did not vary between injured groups, we did find a significant, though modest, increase in the percentage of CD206^+^ macrophages in the group injected with αCD200R1 in comparison with the vehicle one (Figure 4I). The number of neutrophils present in the injured nerve at one day post-injury was evaluated by IHQ in longitudinal sections of the sciatic nerve distally to the crush injury. A significant reduction in neutrophils was found in animals injected with αCD200R1 in comparison with the vehicle group (Figure 4J,K). Next, we wondered whether the reduced infiltration of both monocytes and neutrophils could be due to a reduction in the signaling molecules for their recruitment. Thus, we analyzed the RNA levels of *Ccl2* and *Ccl3*, two of the main chemokines involved in the recruitment of monocytes and neutrophils [28]. QPCR analysis did not show a significant effect of αCD200R1 in the levels of these two chemokines (Figure 4L,M). Myelin clearance deficits are known to impact on successful functional reinnervation. This is a process highly dependent on recruited macrophages and Schwann cells. In order to evaluate whether the reduced macrophage cell recruitment observed after the blockade of the CD200R1 signaling had an impact in the clearance of myelin, longitudinal sections of the lesioned sciatic nerve were made and LFB staining was performed. Gradual clearance of myelin and remyelination was observed over the period studied (1–28 dpi; Figure 5A) with no differences between both injured groups (Figure 5C). Moreover, Oil Red O (ORO) staining was performed at seven days post-injury as an index of neutral lipid droplet accumulation and thus lipid degradation (Figure 5B). Again, no differences in the level of lipid degradation were found between the group injected with αCD200R1 and the vehicle one (Figure 5D).

Finally, we evaluated whether the reduced number of macrophages and their phenotype found at one day post-injury was maintained at seven days post-injury. There were no differences, neither in the number of macrophages nor in their phenotype between groups when analyzed by FACS analysis (Figure 5E,F). Altogether these data suggest that though the inhibition of CD200R1 induces a transient reduction in macrophage recruitment, this does not impact substantially on myelin removal and remyelination and thus it is not the reason behind the impairment of functional reinnervation.

## 4. Discussion

In the present work we show that blocking the signaling through CD200R1 at the time of the nerve lesion reduces macrophages and neutrophils recruitment and impairs long-term functional reinnervation. Surprisingly, while CD200R1 has been shown to deliver an inhibitory and anti-inflammatory signal on microglia and macrophages [15,29], here we observe that CD200R1 signaling inhibition in fact reduces the classical inflammatory response produced after a nerve injury. Blocking CD200R1 does not enhance cytokine/chemokine expression and decreases the number of monocytes recruited early to the lesioned nerve.

CD200R1 has been found to be expressed in the myeloid lineage as well as in lymphocytes in many tissues [15]. However, there are no available data regarding its expression in the PNS. By using QPCR we have detected a constitutive expression of CD200R1 mRNA and protein, and an increase in mRNA expression at all the time points studied after the sciatic nerve crush in comparison with a naive nerve. This result was confirmed by flow cytometry analysis, where we found expression in around 60% of resident macrophages in homeostatic conditions. These data are in accordance with a recent single-cell RNAseq analysis of male and female mice sciatic nerves [30]. After nerve injury, both recruited neutrophils and macrophages expressed CD200R1 in around 10% and 60% of the cells, respectively, though the mean expression of CD200R1 per macrophage in fact was importantly decreased. Thus, this increase in the CD200R1 mRNA expression observed after injury was due to the infiltration of neutrophils and monocytes from the blood rather than to an up-regulation of the receptor.

On the other hand, the PNS CD200 has been found by immunohistochemistry to be expressed preferentially in non-myelinating SCs and specifically into the node of Ranvier and Schmidt–Lanterman structures of myelinating SCs [25]. In the present work, we confirm the expression into the Node of Ranvier of myelinated fibers and in blood vessels, in accordance with other reports showing CD200 expression in endothelial cells [17,19,21,25,30]. The question of why it is expressed in such specific Schwann cell domains, and in which biological functions it is involved, remains to be elucidated. After the lesion, QPCR analysis of CD200 mRNA and protein expression observed by immunohistochemistry revealed a rapid decrease at 24 h, and a later increase, peaking at 14 days post-injury. A decrease in CD200 expression has been shown in other neurological disorders such as spinal cord injury, but also in neurodegenerative disorders, aging and autoimmune diseases [21,31,32,33]. This decrease could be associated either with a downregulation of the protein, due to the death of CD200-expressing neurons, or to the reduction in their neurites associated with these pathologies. In the present study, we hypothesize that the decrease could be due to the axonal breakdown and Schwann cell dedifferentiation, or to a downregulation of CD200 in endothelial cells. In this sense, a work published by Cohen and colleagues found that the late increase in CD200 was associated with newly formed endothelial cells in the epicenter of the lesion after spinal cord injury, and they concluded that CD200-expressing endothelial cells contribute to the downregulation of the inflammatory response and to a more resolving phenotype after injury to the CNS [32]. Further studies need to be performed to clarify the role of CD200 expressed by endothelial cells in regulating inflammation after injuries to the nervous system and whether the expression of CD200 by endothelial cells can modulate the pattern and phenotype of myeloid cells recruitment through this signaling pathway.

Despite the central role of immune receptors in the regulation of the phenotype of innate immune cells, few reports have addressed their expression and role after peripheral nerve injury. Immune receptors such as TREM2 or CD300f are upregulated following the first 3–7 days after a sciatic nerve injury [14,28]. We observed here that CD200R1 follows a similar pattern of increase after the lesion, but somehow more rapidly, as its increase can be detected as early as 24 h after the lesion. However, FACS staining showed that this increase was due to cell infiltration and not to an overexpression in each cell; in fact, each cell expressed lower levels of CD200R1.

In order to unravel why αCD200R1 delayed functional reinnervation, we evaluated different events that take part after crush injury, for instance, the expression of different cytokines and chemokines 24 h after crush injury. After injury to the peripheral nerve, several cytokines and chemokines are upregulated in order to recruit macrophages and neutrophils from the blood [2,28,34]. It has been shown that upregulation of these cytokines is necessary for the accurate influx of neutrophils and monocytes and the further myelin clearance [7,35]. We evaluated the levels of different cytokines and chemokines after the lesion, and though they were increased after the lesion, none of them showed significantly different expression between the group injected with αCD200R1 and the vehicle group. However, when we analyzed the number of macrophages and neutrophils 24 h after crush injury, we found a significant reduction in the number of both cell types in the group treated with αCD200R1 in comparison with the group treated with the vehicle. Although the lower number of macrophages was not maintained at seven days post-injury, it could help explain the further impairment in functional recovery since it has been demonstrated that regeneration is impaired when macrophages are excluded from the site of injury or the infiltration of myeloid cells from blood is somehow impaired [5]. This is in contrast to what has been described for the CNS, where the use of the same αCD200R1 antibody immediately after spinal cord injury had no impact, neither in the entrance of monocytes nor neutrophils [21]. However, in that study macrophages and microglia displayed a more pro-inflammatory phenotype with higher expression of iNOS. In contrast to what was described in the CNS, in the current work we observed a lower entrance of macrophages and neutrophils into the nerve acutely after injury, with a slightly higher expression of macrophage CD206. Most intriguingly, local CD200R1 inhibition by the blocking antibody determined a less inflammatory microenvironment after PNS trauma, in contrast to the pro-inflammatory microenvironment described after CD200R1 inhibition with the same antibody after CNS trauma. Unlike most of the inhibitory immune receptors described, CD200R1 does not present ITIM motifs in its cytoplasmic tail, but three tyrosine residues. After engagement of CD200, CD200R1 associates with DOK1 and DOK2 proteins and recruit SHIP and RasGAP, suppressing the Ras-ERK and PI3K signaling pathways. As a result, there is a downregulation of the expression of pro-inflammatory cytokines [15,36]. With the available data, it is difficult to envision that CD200R1 would convey activating signals. Thus, the disparities in its role in regulating the inflammatory milieu in the PNS and CNS may reflect differences in the cell-types expressing CD200/CD200R1 and their role in inflammation, or in the cell type composition.

Taken together, we hypothesize that in the PNS the CD200 expressed by SCs and endothelial cells is constantly interacting with CD200R1 on resident macrophages that are sensing the micro-environment. After nerve injury, the rapid down-regulation of CD200 by SCs, endothelial cells and putatively exposed axons, and decreased resident macrophage expression of CD200R1 per cell, would lead to a dampened intracellular signaling by CD200R1. This would contribute to mounting the strong inflammatory response necessary for successful functional regeneration. This effect would be reversed at 14 dpi where the increased CD200 expression would potentiate CD200R1 signaling, putatively contributing to resolution. We hypothesize that immediately after the lesion, CD200R1 in resident macrophages generate signals that will increase the recruitment of myeloid cells. This could imply that CD200R1 plays different roles in regulating microglia and macrophage phenotypes, or in the central and peripheral nervous system.

In the present study we assessed regeneration by counting the number of myelinated fibers regenerated at the distal part of the tibial nerve, and reinnervation by counting the number of epidermal fibers reinnervating the skin of the hind-paw. Functional reinnervation, however, was assessed by obtaining the SFI using the Walking track test. The apparently conflicting results showed that, whereas neither the number of regenerated myelinated axons nor the number of epidermal fibers were different between both vehicle and αCD200R1 blocking antibody treated injured groups, the recovery of function measured by the SFI index showed an impairment in the spontaneous functional recovery in animals treated with αCD200R1 blocking antibody. This could be explained by an alteration in axonal conduction. It has been reported that mutations or absence of some molecules required for SC compartmentalization such as periaxin expressed in some SCs domains such as the Cajal Bands modified the internodal length disturbing the nerve conduction velocity, and this was associated with motor and sensory function impairment [37,38]. In this sense, CD200 that is expressed in some SCs domains such as Ranvier Nodes and Schmidt–Lanterman incisures, could have a role in the SCs compartmentalization during development and after nerve injury. Thus, further functional studies need to be performed in order to answer the question of whether CD200 expression is related with internodal length and in this way could affect axonal conduction velocity and functional reinnervation. In wider terms, the question of which is the function of the CD200/CD200R1 pair during normal nerve function, needs to be tackled.

Recent evidence also suggests that CD200R1 can engage ligands other than CD200, such as iSEC1 and iSEC2 [39,40], generating more challenging conditions to unravel the function of this immune receptor. The expression of iSEC1 and iSEC2 were initially described as being restricted to epithelial cells of the digestive tract [39] and just very recently iSEC1 mRNA has been shown to be expressed by sensory neurons of the dorsal root ganglia [40] and in some endothelial cells [30]. It was reported to be involved in the transport of mitochondria from macrophages to neurons [40]. This fact could suggest that problems in the transport of mitochondria by blocking CD200R1 could lead to a failed energy boost needed by axons after nerve injury and thus lead to problems in conduction velocity that could result in the impaired locomotor function recovery. However, it still remains unclear whether iSEC1 protein is expressed in the peripheral nerve by the axons and endothelial cells, and if it has any role through the local communication between axons and myeloid cells such as resident macrophages and recruited blood cells after injury. A single cell atlas of the sciatic nerve under physiological conditions shows that in comparison, CD200 is much highly expressed in the sciatic nerve than iSEC [30]. Thus, it is likely that the effect we observe after blocking CD200R1 is through its CD200 ligand rather than iSEC1, although a deeper study needs to be pursued in order to unravel the role of the different CD200R1 ligands after nerve injury.

## Figures and Tables

**Figure 1 cells-11-01786-f001:**
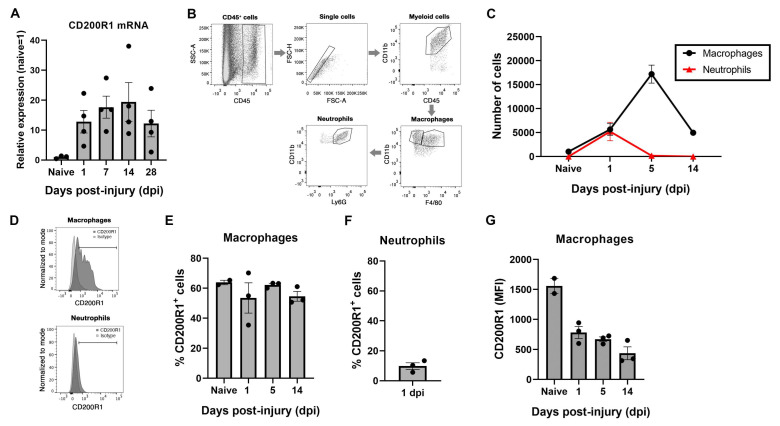
CD200R1 mRNA and protein cell specific expression in naïve conditions and after sciatic nerve crush: (**A**) CD200R1 mRNA expression was observed in naïve nerve and showed an increase from one day post-injury, remaining at high levels in all time points studied; (**B**), Representative FACS dot blots of sciatic nerve crush show the gating for macrophages (CD45^high^, CD11b^high^, F480^+^) and neutrophils (CD45^high^, CD11b^high^, F480^−^, Ly6G^+^) identification; (**C**) graph shows the quantification of the total number of macrophages and neutrophils in the sciatic nerve in naïve and at one, five and fourteen days post-injury; (**D**) representative histograms of the expression of CD200R1 in macrophages and neutrophils at one dpi; (**E**,**F**) graphs show the percentage of macrophages and neutrophils expressing CD200R1 in naïve conditions and at one, five and fourteen days post-injury and at one day post-injury in the case of neutrophils; (**G**) graph shows the median intensity fluorescence (MFI) of CD200R1 expressed by macrophages (*n* = 2–4 animals per time point).

**Figure 2 cells-11-01786-f002:**
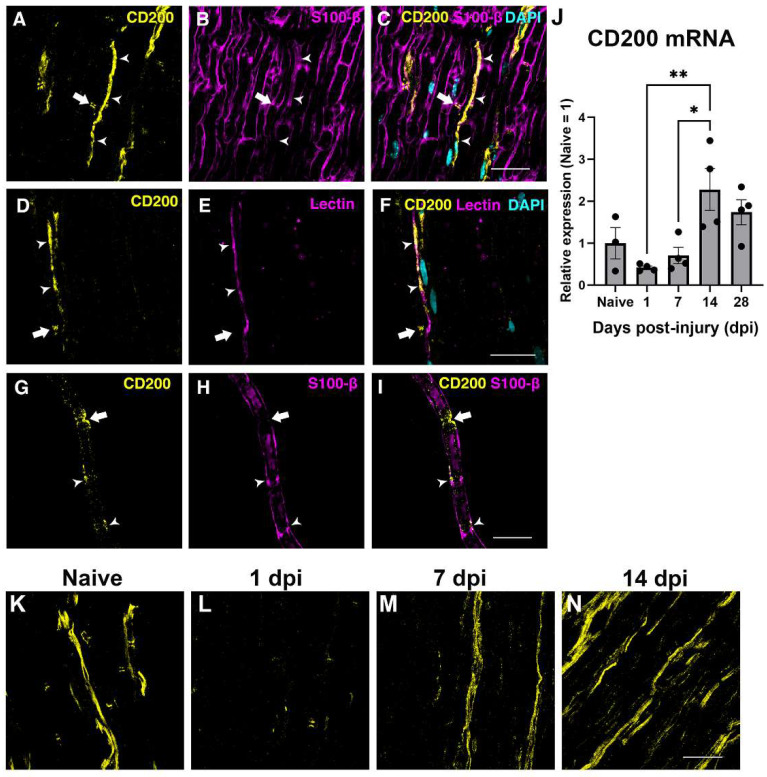
Expression of CD200 in naive and lesioned nerve. Single one micrometer confocal planes of CD200 immunohistochemistry in longitudinal sections of the sciatic nerve show CD200 protein expression in the Ranvier Nodes (arrow) and blood vessels (arrowhead) ((**A**–**F**), S100-B is a Schwann cells marker, and the tomato lectin is an endothelial cell marker). Confocal images of teased fibers of adult sciatic nerve show CD200 expression at the Ranvier Nodes (arrow) and at the Schmidt–Lanterman incisures (arrowheads) (**G**–**I**). CD200 mRNA expression in naive nerve and after sciatic nerve crush injury at different days post-injury (dpi) (**J**) (One-way ANOVA followed by Newman–Keuls post-hoc test * *p* ≤ 0.05; ** *p* ≤ 0.01; *n* = 3–4 animals per time point). Single one micrometer confocal planes of CD200 immunohistochemistry in longitudinal sections of the sciatic nerve in naive conditions and after crush sciatic nerve show a marked reduction in CD200 protein expression at one day post-injury and an increase in the expression from seven days post-injury (**K**–**N**). Scale bars: 30 μm.

**Figure 3 cells-11-01786-f003:**
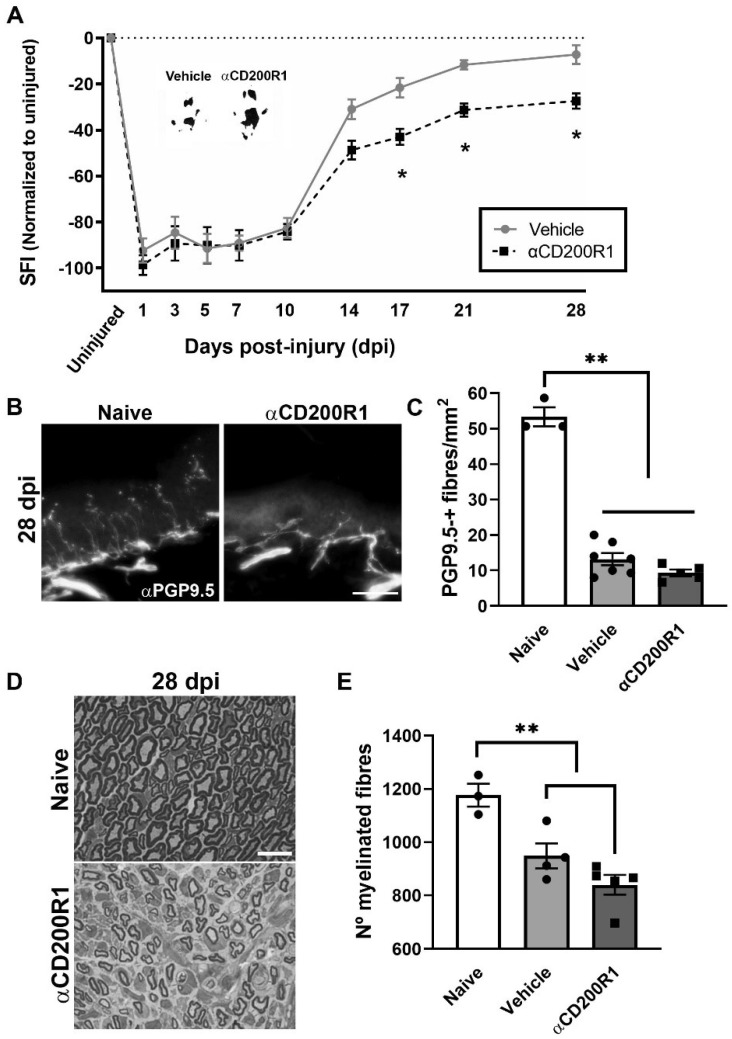
Blockade of CD200R1 impairs functional recovery after crush injury. The Sciatic Functional Index (SFI) walking track analysis reveals an impairment in the functional recovery from 17 dpi until the last point studied (28 dpi) (**A**) (Two-way ANOVA, treatment *p* = 0.002, interaction *p* = 0.03, Bonferroni’s post-hoc test * *p* ≤ 0.05; *n* = 9 vehicle *n* = 8 αCD200R1). Insert shows representative footprints obtained from the vehicle group and the αCD200R1 treated group at 28 dpi. Representative micrographs of plantar skin immunolabeled against PGP9.5 shows the skin innervation in naive and injured αCD200R1 treated mice. Scale bar: 100 µm (**B**). Quantification of the number of intraepidermal nerve fibers at 28 dpi shows a dramatic reduction in the skin innervation after injury in both injured groups in comparison with the naive group (**C**) (One-way ANOVA followed by Tukey’s post-hoc test; ** *p* ≤ 0.0001 compared with naive group). Representative micrograph of transverse sections of the tibial nerve at the ankle level at 28 dpi in a naive and injured mouse treated with αCD200R1. Scale bar: 10 µm (**D**). Quantification of the semithin samples show a reduction in the number of myelinated fibers in both injured groups in comparison to naive (**E**) (One-way ANOVA followed by Tukey’s post-hoc test; ** *p* ≤ 0.001 compared with naive group).

**Figure 4 cells-11-01786-f004:**
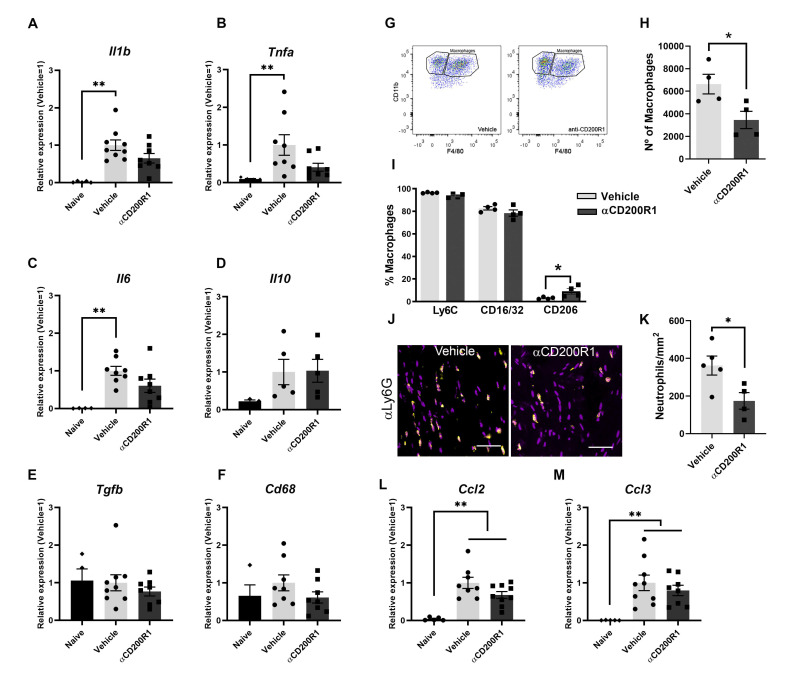
Blockade of CD200R1 reduces monocyte and neutrophils infiltration at 24 h post-injury. QPCR shows a significant increase in several cytokines at one day post-injury with no differences between treatments (**A**–**F**) (Non-parametric Kruskal–Wallis test followed by Dunn’s post-hoc test; ** *p* ≤ 0.01 vs. Naive group). Representative FACS dot blots of crush sciatic nerve show the gating for macrophages (**G**). Quantification of the total number of macrophages at one day post-injury evaluated by FACS analysis and the percentage of macrophages expressing markers of activation as Ly6C, CD16/32 and CD206 (**H**,**I**) (Student t-test; * *p* ≤ 0.05 vs. Vehicle). Representative micrographs of sciatic nerve immunolabeled against Ly6G at 1 dpi (Scale bar: 100 µm) (**J**) and quantification of the number of Ly6G positive cells (**K**). QPCR quantification of the chemokines *Ccl2* and *Ccl3* at 1 dpi (**L**,**M**) (Non-parametric Kruskal–Wallis test followed by Dunn’s post-hoc test; ** *p* ≤ 0.01 vs. Naive group).

**Figure 5 cells-11-01786-f005:**
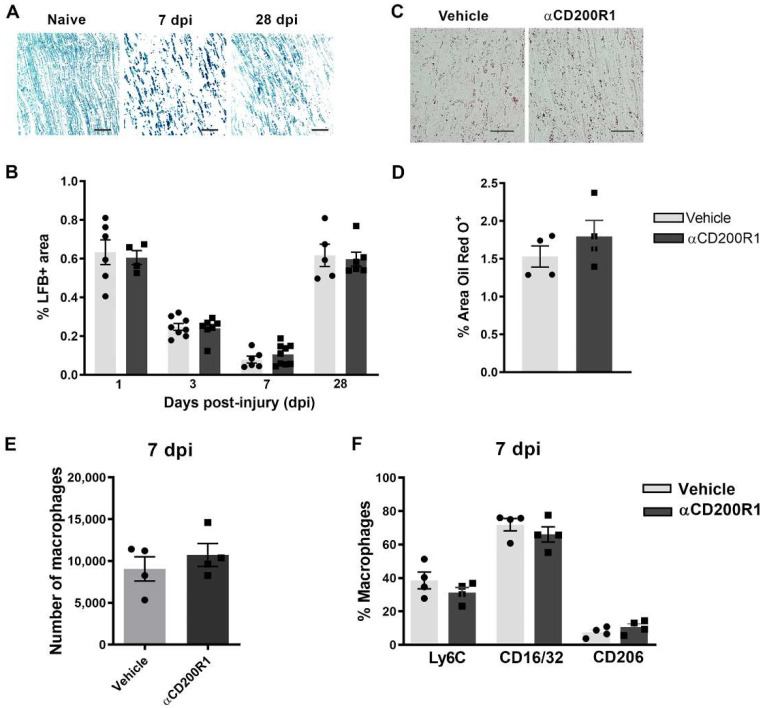
Acute blockade of CD200R1 does not affect myelin clearance after crush injury. Representative images of Luxol Fast Blue (LFB) staining in naive nerve and after crush injury at seven and 28 dpi. Scale bar: 50 µm (**A**). Graph shows LFB staining represented as a percentage of area stained (**B**). Representative images of Oil Red O (ORO) lipid droplet accumulation at seven days post-crush in animals treated with vehicle and anti-CD200R (Scale bar: 50 µm) (**C**) and quantification of the area stained with Oil Red O represented as a percentage of area stained (**D**). Quantification of the total number of macrophages at seven days post-injury (**E**) evaluated by FACS analysis and the percentage of macrophages expressing markers of activation as Ly6C, CD16/32 and CD206 (**F**).

## Data Availability

All data generated in the current study are available from the corresponding author on reasonable request.

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
