# Peer review of "CD200R1 Contributes to Successful Functional Reinnervation after a Sciatic Nerve Injury"

_cells, 2022, doi:10.3390/cells11111786_

Round 1
Reviewer 1 Report
When reading this paper, l was initially very excited at the findings but l have now a serious concern that has to be raised.
1) I made the assumption, and l think the authors did to, that the antibody treatment would block CD200R signaling and thus exacerbate inflammation. This clearly did not happen. The results for the cytokines suggest a decrease but in practice might be no effect (according to the statistics). My concern is that the antibody to CD200R might actually be activating signaling not blocking it. This would explain the biological effects observed even if the cytokines were not changed. Depending on the epitopes of this antibody, receptor activation can occur. There has been precedent for other antibodies.
So, l would suggest that the authors provide evidence concerning the specificity of this rat monoclonal. It appears to be available from multiple commercial suppliers.
a) I would ask the authors to provide published or laboratory data that this antibody actually does block the receptor signaling. This might need to be done in vitro or proper reference to previous published work. An in vitro experiment using even macrophages would be sufficient. This experiment would also confirm whether (as l suspect) that there is some activation of CD200R signaling. It seems strange to me that some effects were observed so quickly. I doubt that this was due to preventing CD200 binding.
b) I would suggest that the authors make the introduction and discussion more concise. They are well written on the whole but are too long and makes the paper less readable. A lot of it can be kept for review articles. What was missing was succinct discussion of what is known about CD200R signaling and therefore how it might explain the observations.
c) The decrease and then increase in CD200 expression was interesting. How this explains the mechanisms being presented could be integrated into the study.
d) Do you have any western blot or IHC data on expression of CD200R using this antibody.
e) A little bit of English editing is needed to increase clarity. I suggest shortening some of the long sentences.
If you can prove that there is actual blockade of CD200R, this will be a significant paper. If it is activation, l still think it would be a significant paper but some rewriting is needed. CD200/CD200R has not had the attention it deserves so l encourage the authors to address this issue. The cytokine data might be significant if more animals were studied, or might just reflect normal variability.
Author Response
Reviewer #1:
When reading this paper, l was initially very excited at the findings but l have now a serious concern that has to be raised.
1) I made the assumption, and l think the authors did to, that the antibody treatment would block CD200R signaling and thus exacerbate inflammation. This clearly did not happen. The results for the cytokines suggest a decrease but in practice might be no effect (according to the statistics). My concern is that the antibody to CD200R might actually be activating signaling not blocking it. This would explain the biological effects observed even if the cytokines were not changed. Depending on the epitopes of this antibody, receptor activation can occur. There has been precedent for other antibodies.
So, l would suggest that the authors provide evidence concerning the specificity of this rat monoclonal. It appears to be available from multiple commercial suppliers.
a) I would ask the authors to provide published or laboratory data that this antibody actually does block the receptor signaling. This might need to be done in vitro or proper reference to previous published work. An in vitro experiment using even macrophages would be sufficient. This experiment would also confirm whether (as l suspect) that there is some activation of CD200R signaling. It seems strange to me that some effects were observed so quickly. I doubt that this was due to preventing CD200 binding.
Response: We were also surprised, as reviewer #1, that the treatment with the anti-CD200R1 blocking antibody induced a decreased inflammatory milieu instead of inducing inflammation. The OX131 antibody against CD200R1 that we have used in the current work, has been characterized [1]. It is monoclonal and recombinant, and thus is more reproducibly produced, limiting batch variations. It has also been engineered to minimize FcγR receptor binding and has also showed reduced complement binding by Absolute Antibody, the supplier of the antibody (https://absoluteantibody.com/). The OX131 αCD200R1 used in our study was originally generated by the group of Neil Barclay and has been shown to be specific for CD200R1 in C57BL/6 mice (this antibody also recognizes CD200RLe receptor which is not expressed in C57BL/6 mice, but does not recognize CD200RLa, CD200RLb and CD200Rc, which are found in this mouse strain) [1]. The authors observed by surface plasmon resonance that OX131 mAb blocked CD200 binding to CD200R1 and that OX131 mAb interaction with CD200R1 prevented inhibitory signaling induced by CD200. In the functional studies performed in the work by Akkaya et al, OX131 also gave week agonistic inhibitory signals when engaged to CD200R1, however, in the presence of CD200 (as it is the case for sciatic nerve) its net effects were clearly blocking engagement of CD200R1 by CD200.
Additional evidence suggesting that in vivo OX131 has a CD200R1 blocking effect comes from another work published by our group [2]. Here, the OX131 mAb from the same supplier was used to study the role of the pair CD200-CD200R1 in spinal cord injury. By injecting OX131 mAb into the spinal cord parenchyma immediately after a contusion to the spinal cord (in a very similar protocol used in the sciatic nerve injury study presented here), we found an impairment in spontaneous locomotor recovery associated with the induction of a pro-inflammatory phenotype. By contrast, when injecting a recombinant CD200 we found a significant improvement in functional recovery. With these results in mind, and taking into account that the OX131 antibody was proinflammatory in vivo after a spinal cord injury as expected, we strongly believe that OX131 mAb is acting mainly as a blocking antibody in the presence of CD200.
This has been discussed in the text for further clarification of this point (lines 323-326).
Additionally, and due to the reviewer’s concerns, we have performed a small experiment with bone marrow derived macrophages (BMDMs) from CD200R1-knockout and wildtype mice treated with LPS and OX131 anti-CD200R1 and analysed the level of mRNA for IL1β by QPCR. Briefly, we tested the cytokine expression 24 hours after different conditions such as no treatment (basal), anti-CD200R1 OX131 mAb (5 μg/ml), LPS (500 ng/ml) or LPS + anti-CD200R1 OX131 mAb. As expected, the results show that both Wildtype and CD200R1 knockout BMDMs increases cytokine production in response to LPS, being this effect increased in KO macrophages in comparison with WT. Treatment with OX131 anti-CD200R1 increased Il1b mRNA expression in WT macrophages with LPS, but not in KO macrophages, suggesting that the antibody is indeed inhibiting CD200R1 function rather than activating the receptor.
b) I would suggest that the authors make the introduction and discussion more concise. They are well written on the whole but are too long and makes the paper less readable. A lot of it can be kept for review articles. What was missing was succinct discussion of what is known about CD200R signaling and therefore how it might explain the observations.
Response: As suggested, the introduction and discussion has been shortened, for example lines 61 to 64 and lines 92-93 in the introduction and lines 480 to 494 in the discussion have been deleted. Furthermore, we have added some discussion about CD200R signaling (lines 529-539).
c) The decrease and then increase in CD200 expression was interesting. How this explains the mechanisms being presented could be integrated into the study.
Response: We believe that after shortening the manuscript, the discussion is now more focused, and this point becomes highlighted.
d) Do you have any western blot or IHC data on expression of CD200R using this antibody.
Response: This antibody appears not to be very good after fixation of the samples, and thus we have used another well-characterized anti-CD200R1 antibody for protein detection.
e) A little bit of English editing is needed to increase clarity. I suggest shortening some of the long sentences.
Response: As suggested, some English editing has been performed.
If you can prove that there is actual blockade of CD200R, this will be a significant paper. If it is activation, l still think it would be a significant paper but some rewriting is needed. CD200/CD200R has not had the attention it deserves so l encourage the authors to address this issue. The cytokine data might be significant if more animals were studied, or might just reflect normal variability.
Response: In order to address this issue, we performed an additional experiment with crush injury + anti-CD200R1 or Vehicle. We added 4 animals per each treated group and 3 naïve animals. mRNA extraction, retrotranscription and Q-PCR have been done with all the animals and results have been re-analyzed. Figure 4 and its figure legend have been changed adding more animals and stats to the graph. However, even after increasing the number of animals, we did not find significant differences between both treated groups (anti-CD200R1 versus Vehicle treated group), suggesting that this reflects the actual biological effects.
Reviewer 2 Report
- Please provide the approval number received from the Institut Pasteur de Montevideo Animal Care Committee (line 104).
- Please provide the reason why female mice were used for flow cytometry analyses only.
- Please provide the name of city, country for reagents/antibodies used in the current study.
- In Figure 1A, there are large individual differences of CD200R1 mRNA. It could be the result of small sample size or incorrect annealing temperature. The authors need to re-conduct the qPCR.
- In Figure 2, figure labels need to be edited. 'G' in the CD200mRNA graph should be replaced with 'J'. 'H-K' at the bottom of the figure should be replaced with 'K-N'.
- The authors need to provide the validation data that confirms the reduction of CD200R1 protein level in the aCD200R1-treated mice compared with vehicle treated mice via Immunostainining, ELISA, or Western blotting.
- In figure 4, there were no difference between naive and vehicle groups in inflammatory mediator levels at 1dpi. What about 7 dpi?
Author Response
Reviewer #2:
1. Please provide the approval number received from the Institut Pasteur de Montevideo Animal Care Committee (line 104).
Response: The approval number, 005-19, has been included in the manuscript (line 104)
2. Please provide the reason why female mice were used for flow cytometry analyses only.
Response: In the latest years there is awareness of the need to include sex as a biological variable. In the case of studies involving inflammation pathways, it is increasingly more evident that inflammatory response after different insults to tissues can be different depending on the sex. Although sex differences have been described for other immune receptor function, none have been reported yet for CD200R1. Moreover, unpublished results from colleagues of the lab show that rates of nerve regeneration are not affected by sex (manuscript currently under second revision). Furthermore, a work published by our group studying the involvement of another immune receptor, CD300f, in peripheral nerve regeneration, we used both males and females mice in mixed groups and we did not find differences in the rate of regeneration due to sex [3]. In the current report, and in order to apply the 3Rs to our research, we used female mice for FACS as the males were used for regeneration studies. Nevertheless, this is an interesting observation that needs further study in the future and that it was beyond our scope in the current work.
3. Please provide the name of city, country for reagents/antibodies used in the current study.
Response: We have included all the information required by the reviewer.
4. In Figure 1A, there are large individual differences of CD200R1 mRNA. It could be the result of small sample size or incorrect annealing temperature. The authors need to re-conduct the qPCR
Response: Quantitative PCR was performed using Taqman probes which are highly specific and well tested. We were also surprised by the variability found at the different time points studied after nerve injury. However, in naïve animals we do not see such variability and thus we consider that the variability found is because of the variability of the lesion rather than the qPCR technique.
5. In Figure 2, figure labels need to be edited. 'G' in the CD200mRNA graph should be replaced with 'J'. 'H-K' at the bottom of the figure should be replaced with 'K-N'.
Response: We have already modified the figure.
6. The authors need to provide the validation data that confirms the reduction of CD200R1 protein level in the aCD200R1-treated mice compared with vehicle treated mice via Immunostainining, ELISA, or Western blotting.
Response: At this point, we think that it has been a misunderstanding as we did not evaluate CD200R1 protein levels in animals treated with aCD200R1. Moreover, we did not analyse CD200R1 protein levels after treatment with aCD200R1 as we do not expect any changes in CD200R1 expression after the treatment.
7. In figure 4, there were no difference between naive and vehicle groups in inflammatory mediator levels at 1dpi. What about 7 dpi?
Response: Thank you for the comment. When we analysed the results of the QPCR of inflammatory mediators, we focused on the differences between the injected groups and we did not consider to add the stats of the naïve group.
We have performed an additional experiment with crush injury + anti-CD200R1 or Vehicle. We have done 4 animals per each treated group and 3 naïve animals. mRNA extraction, retrotranscription and Q-PCR have been done with all the animals and results have been analyzed. Figure 4 and its figure legend have been changed adding more animals and stats to the graph. Even after increasing the number of animals, we did not find significant differences between both treated groups (anti-CD200R1 versus Vehicle treated group), suggesting that this reflects the actual biological effects.
Moreover, since you commented this fact, we re-analysed the statistically and we found differences between naïve and injured group treated with the vehicle in some of the cytokines analysed such Il1b, TNFa, IL6 and the chemokines CCL2 and CCL3 where both injured groups were significantly different with the naïve group. We have changed the figure 4 adding the stats. For the second question about what happens at 7 dpi, we did not study the inflammatory mediators at 7 dpi as we were not expecting differences in the levels of these cytokines and chemokines between naïve and injured groups so late in time due to that only one injection of the antiCD200R1 antibody. Moreover, it has been previously shown that levels of cytokines and chemokines increase rapidly after injury and then diminish in the next few days [4,5], and thus we did not contemplate to look at this time point.
1. Akkaya, M.; Aknin, M.L.; Akkaya, B.; Barclay, A.N. Dissection of Agonistic and Blocking Effects of CD200 Receptor Antibodies. PLoS One 2013, 8, doi:10.1371/journal.pone.0063325.
2. Lago, N.; Pannunzio, B.; Amo-Aparicio, J.; López-Vales, R.; Peluffo, H. CD200 Modulates Spinal Cord Injury Neuroinflammation and Outcome through CD200R1. Brain. Behav. Immun. 2018, 73, 416–426, doi:10.1016/j.bbi.2018.06.002.
3. Peluffo, H.; Solari-Saquieres, P.; Negro-Demontel, M.L.; Francos-Quijorna, I.; Navarro, X.; López-Vales, R.; Sayós, J.; Lago, N. CD300f Immunoreceptor Contributes to Peripheral Nerve Regeneration by the Modulation of Macrophage Inflammatory Phenotype. J. Neuroinflammation 2015, 12, 1–15, doi:10.1186/s12974-015-0364-y.
4. Rotshenker, S. Wallerian Degeneration : The Innate-Immune Response to Traumatic Nerve Injury. J. Neuroinflammation 2011, 8, 109, doi:10.1186/1742-2094-8-109.
5. Shamash, S.; Reichert, F.; Rotshenker, S. The Cytokine Network of Wallerian Degeneration: Tumor Necrosis Factor-α, Interleukin-1α, and Interleukin-1β. J. Neurosci. 2002, 22, 3052–3060, doi:10.1523/jneurosci.22-08-03052.2002.
Round 2
Reviewer 1 Report
The authors have addressed the issues l raised very thoroughly. They have addressed the issues l had about the antibody specificity. Although it is not trendy, l would always like to see some experimental data showing the specificity of any antibody in the authors' hands. It is not uncommon to have batch to batch manufacturing issues of the same clone.
The additional in vitro validation experiment l asked for and was carried out should be included in supplemental data.
Overall, though l think this paper adds significant content to studies of CD200 and CD200R1 in vivo. The results were not as l would have expected so this makes it more interesting.
The journal does ask to assess "interest to readers" so l would again suggest that the authors see if they can shorten the discussion. We are in a "TLDR" situation (too long did not read) for many of the potential readers.
Author Response
Thank you for the comments and suggestions.
We have added the figure of BMDM experiment as a supplementary figure 1 and the results between lines 339-341. Moreover, we added the information about CD200R1-KO mice used in the experiment and the BMDM cell culture in the material and methods section (lines 107-110 and lines 243-255, respectively).